# Ten “Cheat Codes” for Measuring Oxidative Stress in Humans

**DOI:** 10.3390/antiox13070877

**Published:** 2024-07-22

**Authors:** James N. Cobley, Nikos V. Margaritelis, Panagiotis N. Chatzinikolaou, Michalis G. Nikolaidis, Gareth W. Davison

**Affiliations:** 1The University of Dundee, Dundee DD1 4HN, UK; 2Ulster University, Belfast BT15 1ED, Northern Ireland, UK; gw.davison@ulster.ac.uk; 3Aristotle University of Thessaloniki, 62122 Serres, Greece; nvmargar@phed-sr.auth.gr (N.V.M.); chatzinpn@phed-sr.auth.gr (P.N.C.); nikolaidis@auth.gr (M.G.N.)

**Keywords:** oxidative stress, ROS, oxidative damage, redox regulation, antioxidant, method

## Abstract

Formidable and often seemingly insurmountable conceptual, technical, and methodological challenges hamper the measurement of oxidative stress in humans. For instance, fraught and flawed methods, such as the thiobarbituric acid reactive substances assay kits for lipid peroxidation, rate-limit progress. To advance translational redox research, we present ten comprehensive “cheat codes” for measuring oxidative stress in humans. The cheat codes include analytical approaches to assess reactive oxygen species, antioxidants, oxidative damage, and redox regulation. They provide essential conceptual, technical, and methodological information inclusive of curated “do” and “don’t” guidelines. Given the biochemical complexity of oxidative stress, we present a research question-grounded decision tree guide for selecting the most appropriate cheat code(s) to implement in a prospective human experiment. Worked examples demonstrate the benefits of the decision tree-based cheat code selection tool. The ten cheat codes define an invaluable resource for measuring oxidative stress in humans.

## 1. Introduction

Many researchers investigating human physiology in health and disease need to measure oxidative stress. For example, one must measure oxidative stress to determine whether a nutritional intervention, such as vitamin E [1], acts as an antioxidant to regulate exercise adaptations [2,3]. However, formidable conceptual, technical, and methodological challenges pose significant barriers to measuring oxidative stress in humans [4,5]. These challenges, which include the inapplicability of cutting-edge genetically encoded reactive oxygen species (ROS) probes in human studies [6,7] and the daunting task of choosing among fraught and flawed assays, such as the thiobarbituric acid reactive substances (TBARS) assay [8,9], rate-limit progress. They even deter many researchers from measuring oxidative stress altogether. This predicament can be likened to a novice navigating a challenging level in a video game, where “cheat codes”—shortcuts and/or strategies to overcome difficult obstacles—would help solve complex, seemingly intractable, problems. To assist those grappling with persistent methodological challenges, we define key terms, such as ROS, and elaborate ten comprehensive “cheat codes” for measuring oxidative stress in humans. Finally, we present a research question-grounded decision tree guide for selecting the most appropriate cheat code(s) to implement in a prospective human experiment before offering concluding perspectives. To illustrate how the cheat codes can be applied to study oxidative stress in humans, we provide a consistent set of examples from the exercise redox biochemistry literature.


**Part 1. A brief guide to oxygen, ROS, antioxidants, and oxidative stress**


### 1.1. Oxygen

Our 37 trillion cells vociferously consume around 3.5 millilitres per minute of a free radical: oxygen. Diatomic oxygen contains two unpaired electrons: it is a diradical [10]. The lone, unpaired electrons in oxygen spin in parallel antibonding orbitals [11] (e.g., ↑↑; see Figure 1). Aerobic life is possible because oxygen cannot react with the vast majority of electron spin-paired molecules in the body due to the reactions being spin-forbidden [12]. To explain spin-forbidden, think of the card game “snap”. Our goal is to “snap”, that is, react, our cards, but we can only “snap” spin-allowed matches. If you play an oxygen card and we play a guanine DNA base card, then we cannot “snap” as a result of the following:

No appreciable reaction = oxygen: ↑↑ guanine-DNA ↓↑ (spin violated).

If you could “flip” the spin, as it happens when UV light excites oxygen to yield non-radical singlet oxygen [13], then the first one to “snap” wins the game owing to the following:

Appreciable reaction = singlet oxygen ↓↑ guanine-DNA ↓↑ (spin allowed) [14].

### 1.2. ROS

Oxygen appreciably reacts with molecules able to donate one electron, termed univalent reduction (see Equation (1)), leading to a zoo-like menagerie of free radical (e.g., superoxide) and non-radical (e.g., hydrogen peroxide) ROS (see Figure 1). As an insightful account remarked [15], ROS was first used in the abstract of a paper in 1977 [16]. Before “ROS” became *de rigueur*, virtually every paper named the specific species being studied (e.g., [17]). Many problems stemmed from failing to state the specific species [18] and certain misconceptions about “ROS” leading to interpretational errors. Selected instructive points about ROS include:Superoxide is not necessarily super. The name “superoxide” originated from the odd stoichiometry of a chemical reaction in 1934 [19]. It had nothing to do with any special “super” biochemical reactivity as an oxidant [20]. Sawyer and Valentine commented that the probability of superoxide oxidising a molecule to yield the peroxide dianion is nil. Moreover, McCord and Fridovich discovered superoxide dismutase (SOD) by observing that superoxide reduced ferric cytochrome c [21,22].Each ROS is biochemically unique. Superoxide appreciably reacts with a small number of targets, such as tryptophan free radicals [23]. Conversely, the ferocious hydroxyl radical rapidly reacts with virtually every organic molecule at a diffusion-controlled rate [24].There is no set percentage rate of superoxide production from consumed oxygen [25,26]. As studies comparing rest with exercise attest [27], the dynamic variable rate of mitochondrial superoxide production is context-dependent [28,29].To quote Sies and Jones, “*ROS is a term, not a molecule*” [30].
Oxygen + electron → superoxide(1)

### 1.3. Antioxidants

Even luminaries struggle to classify “antioxidants” [31,32]. Unexpected, class-defying antioxidants include multicellularity as a defence against oxidative stress by limiting oxygen exposure [33], uncoupling proteins to decrease the probability of mitochondrial superoxide production by lowering the electrochemical proton motive force [34,35] and the diverse proteins responsible for repairing oxidative damage [36]. Despite the perennial search for the best catch-all definition, experts [37] define an antioxidant as follows:

“*any substance that delays, prevents or removes oxidative damage to a target molecule*”

                  OR

“*a substance that reacts with an oxidant to regulate its reactions with other targets, thus influencing redox-dependent biological signalling pathways and/or oxidative damage*”.

A protein repairing oxidised DNA would fall into the first category, whereas catalase (CAT) would fall into the second. Selected instructive points about antioxidants include:There is no individual “best” antioxidant. Just like ROS, they are all different (see Figure 2).A few “frontline” enzymes like SOD perform most of the redox “heavy lifting”. Ordinarily, SOD isoforms consume most of the superoxide produced in a cell [38]. So, only picomoles remain for other molecules, such as vitamin C, to “scavenge”. For example, 3.01 × 10^18^ (5 µM) superoxide molecules can be produced per second in *Escherichia coli*, but SOD limits [superoxide] by 4-logs to 6.0 × 10^14^ molecules corresponding to 0.0009 µM or 900 picomoles [39].An antioxidant is context-dependent. SOD generates hydrogen peroxide and oxygen. So, it is an antioxidant in concert with other enzymes [40]. Further, when mis-metalled, the mitochondrial SOD isoform produces hydroxyl radical [41].Many antioxidants “moonlight”. Sticking with SOD, the copper zinc isoform can act as a transcription factor [42] and a cysteine oxidase [43]. Relatedly, SOD regulates electrochemistry by metabolising the negatively charged superoxide anion, an excellent Brønsted base [44].Some antioxidant enzymes react with many species. Like superoxide also reacts with (and inactivates) CAT and glutathione peroxidase (GPX) [45,46], emerging evidence demonstrates that SOD metabolizes hydrogen sulphide [47], which complements prior evidence of reactivity, albeit kinetically slow, with hydrogen peroxide [48].No specific antioxidants target the hydroxyl radical. If you had a 100 kg athlete, you would need to load them with 50 kg of the antioxidant to meaningfully scavenge the hydroxyl radical on mass action and kinetic grounds [49,50].Reactive species can be antioxidants. Nitric oxide reacts with lipid peroxyl radicals to terminate a free radical chain reaction [51,52]. The nitrated lipids so-formed may possess anti-inflammatory properties [53].As in real estate, location matters. Polyphenols may be antioxidants in the gut [54] before being metabolised to and secreted in pro-oxidant forms [55]. Touted “antioxidants” like polyphenols often exert beneficial effects by acting as pro-oxidants [56,57].

### 1.4. Oxidative Stress

Sies [58] defined oxidative stress as an imbalance between ROS and antioxidants (AOX) in favour of the former that leads to oxidative damage (Equation (2)). Oxidative damage includes chemically diverse oxidised lipid, DNA/RNA, and protein molecules. For example, 2-oxohistidine is an oxidised form of the amino acid histidine [59]. While there is always some oxidative damage, persistent and excessive levels of unrepaired molecules are linked to certain diseases [60], most notably cancer [61,62,63]. In response to mounting evidence [64,65,66], Sies and others redefined oxidative stress to include redox regulation [67,68,69,70,71] (Equation (3)).
ROS > AOX = ↑ oxidative damage. (2)
↑ROS + ~AOX = redox regulation. (3)

Variations in Equations (2) and (3) are possible, such as ↑ oxidative damage = ~ROS + ↓AOX. The equations formulate input–output relationships (see Figure 3). The biochemical reactions underlying the output constitute a “process” module (input → process → output) [72]. For example, Equation (1) specifies an ROS input, such as increased nuclear hydroxyl radical generation, producing an oxidative damage output, such as guanine base oxidation in DNA, via process-specific biochemical reactions [73]. In this case, the AOX module would comprise factors that constrain the Fenton reaction (see Equation (4)) inclusive of hydrogen peroxide availability as influenced by proximal superoxide production, the nature of the iron ligand, and enzymes, such as CAT. Like many things in redox biology, the identity of the oxidising species formed by the Fenton reaction, first described in 1876, is still contested [74,75,76].
H_2_O_2_ + Fe^2+^ → OH^−^ + ·OH + Fe^3+^ (this equation is an example of the many possible reactions in Fenton systems)(4)

Is oxidative stress good, bad, or neither? The answer, and the source of much confusion, is all three. Whether biochemical reactions are “good” or “bad”, in so far as anything can be so simply classed, depends on the context. Context-dependent functionality means different interpretations of the same immutable biochemistry can coexist. Like a picture, the same oxidative stress frame can contain “good” and “bad” pixels whereby each pixel defines specific biochemical reactions. The pixels illuminated by a functional measurement spotlight can lead to different and even diametrically opposed interpretations of the same redox picture [77]. Measurements influence interpretations. In humans, whole-body metrics, like peak power output, vs. molecular proxies, like mRNA levels, lead to a neutral vs. negative effect of the same redox picture in the context of antioxidants influencing exercise adaptations [78].

Oxidative stress is evolving to a mobile element fluctuating about a spectrum bookended by “good” oxidative eustress and “bad” oxidative distress [79,80,81,82]. Like the goldilocks homeostatic principles, the modern view that cells need to produce some ROS in light of their beneficial effects implies that insufficient levels of ROS can be harmful, termed reductive stress. Despite misgivings about the term [83], recent evidence unearthed a transcriptional response to insufficient ROS levels [84,85]. To help researchers clearly communicate their work, we recommend that they define key terms by using suggested and in some cases “operational” definitions (see Table 1).


**Part 2. Ten “cheat codes” for measuring oxidative stress in humans**



**Cheat code 1. Avoid the minefield of measuring ROS directly in humans (at least for now)**


Although forward-looking possibilities may permit their measurement in the future (hence the “at least for now” clause), it is exceptionally difficult to measure ROS in humans because:One invariably exposes the sample to 21% oxygen. Mitochondrial superoxide production depends, in part, on [oxygen] [86]. So, raising [oxygen] from 1–10% to 21% [87] by aerating the sample would be expected to artificially increase superoxide production [88].The “ROS” in the sample will have inevitably disappeared before one can measure them. They ephemerally flit in and out of existence on nanosecond timescales (10^−9^ of a second). So, what is one really measuring? Potentially, the rate of artificial ROS production in a heavily oxygenated sample [37].Although it is possible to minimise the above (e.g., degassing the sample and rapidly adding a probe), it is arduous. Even if artificial generation were minimised, a superoxide probe, for example, must still compete with SOD [89], hampering the ability to detect all of the molecules in the sample. There is always the possibility of inadvertent artefacts, such as the release of redox-active iron ions from the haemolysis of erythrocytes in blood samples.Many lysis buffers contain ROS, such as hydrogen peroxide and lipid hydroperoxides (LOOH) [90].Cutting-edge genetically encoded probes cannot be used in humans [91].

Fortunately, situations, such as the classic studies demonstrating that exercised skeletal muscle tissues produce free radicals [92], where one needs to measure ROS in humans are relatively rare. Nevertheless, researchers seeking to measure ROS may wish to (1) avoid the fluorescent probe 2′,7′-dichlorodihydrofluorescein, as it produces ROS [93,94], and kits claiming to exclusively measure the hydroxyl radical; (2) carefully read assay kits; and (3) implement a valid technique, such as electron paramagnetic resonance-based spin trapping [95]. Even amongst experts routinely measuring ROS, unmet methodological needs perennially inspire new approaches [96], such as the evolution from pH-sensitive to -insensitive hydrogen peroxide sensors [6] or boronic to borinic acid-based fluorescent probes for kinetically more efficient hydrogen peroxide sensing [97]. Measuring ROS does not reveal what they are doing, which is ultimately what one wants to discover [9].


**Cheat code 2. How to infer ROS production in human samples by using endogenous reporter molecules**


Inspired by the Kalyananraman group [98], one can infer ROS production in humans (see Figure 4) by using endogenous reporter molecules of free radicals (aconitase) and non-radicals (peroxiredoxins (PRDX)). First, superoxide and other selected species, such as peroxynitrite, react exceptionally fast with aconitase [99]. Peroxynitrite production depends on superoxide: nitric oxide + superoxide = peroxynitrite [100]. By depriving it of an essential iron atom, they inactivate aconitase [39]. By measuring aconitase activity, as detailed elsewhere [101,102,103], it is possible to obtain a surrogate readout of the levels of selected ROS. Unlike ectopic approaches whereby adding a probe may perturb the biological system [104], endogenous reporters preserve the natural state.

Second, hydrogen peroxide and species with an O-O bond like peroxynitrite and LOOH [105] react fast with PRDX isoforms [106,107,108]. Their reaction with the catalytic cysteine of most PRDX isoforms produces a sulphenic acid, which condenses with a neighbouring cysteine to form a disulphide crosslink [109,110]. The crosslink covalently staples two monomers together, forming a dimer. Sometimes, hydrogen peroxide reacts with the sulphenic acid to form an “over-oxidised” sulphinic acid [111,112], with further reactions yielding sulphonic acids [113]. One can readily measure PRDX1-3 oxidation by non-reducing immunoblot, the so-called dimer assay, and the over-oxidation of multiple isoforms by conventional immunoblotting [114].

These techniques have been used to great effect in humans. For example, the Jackson group deployed them to infer exercise-induced ROS production in human skeletal muscle [115,116]. These assays provide an integrated readout of the activity of a key antioxidant enzyme and the levels of ROS, especially when combined with similar assays for the thioredoxin redox system [117]. The ubiquity of PRDX enzymes means the assays can be applied systemically and in tissues [118]. The amount of PRDX2 [119] in erythrocytes may support fingertip-based oxidative stress testing.


**Cheat code 3. How to hack “TAC” in human samples**


As Sies remarked [70], the term total antioxidant capacity (TAC) originated from the ability of plasma to handle an ectopic redox stimulus: azo-initiator-induced peroxyl radical production [120]. Given the lack of most antioxidant enzymes (in a redox-active form) in plasma [31], the assay can test non-enzymatic antioxidant capacity (NEAC) against specific free radicals in plasma [121]. If TAC is used judiciously, it is possible to gain some albeit limited insights. To hack TAC in humans, consider that:It is an artificial redox challenge imposed on ex vivo biological material and may have questionable relevance to the ability of said material to “defend” against other species, such as superoxide.It can be useful with aqueous antioxidants, like vitamin C, in so far as confirming nutrient loading, when combined with assays to measure the nutrient content, and potential for redox-activity. The potential is non-equivalent to the actual activity.The actual antioxidant activity of blood plasma is influenced by erythrocytes and surrounding tissues, such as the endothelium.There are many commercial “kits” for TAC. Please carefully consider their use and properly report their information. Statements like “*TAC was measured with X-kit*” without detailing the procedure are discouraged.Use other assays to better interpret TAC in plasma (see cheat codes 4–9) and refrain from measuring it in tissues; as a general rule one would be better advised to measure antioxidant enzymes.Low-molecular-weight “antioxidants” also contribute to TAC activity. For example, peroxyl radicals react fast with cysteine, such as the micromole levels of albumin cysteine in plasma. Free radical chain reactions generate other ROS, such as superoxide [40,122].The word total, unless carefully qualified (as in non-enzymatic capacity against peroxyl radical), is a misleading misnomer [123].


**Cheat code 4. How to measure antioxidants in human samples**


As a sizeable exercise literature study reviewed elsewhere attests [124,125,126], antioxidants can be measured in human samples using established methods (see Table 2).

Curated “do” and “don’t” guidelines for measuring antioxidants in human samples include:Do consider the assay biochemistry. For instance, some SOD assays are prone to artefacts arising from other molecules able to reduce ferric cytochrome c and the complex biochemistry of assay reporter molecules, such as nitro-blue tetrazolium [141].Do quantify the systemic release of antioxidant enzymes by ELISA and immunoblot [142], especially in exosomes [143]. Do not measure GSH or antioxidant enzyme activity (e.g., GPX) in plasma/serum [4]. The concentrations of GSH, glutathione reductase, and NADPH needed to sustain appreciable plasma GPX activity are minimal.Do use HPLC, with appropriate controls to block artificial oxidation (e.g., *N*-ethylmaleimide or iodoacetamide [144]), to quantify GSH and GSSG. Do not read too much into the thermodynamic reducing potential of the redox couple (GSH/GSSG) as computed using the Nernst equation [145].Do follow best practice for quantitative immunoblotting using validated antibodies [146,147].Do consider the possibility that enzyme activities measured ex vivo may not reflect what is possible in vivo. For example, for thioredoxin reductase, much would depend on the continual supply of NAPDH [148].Do consider that there is no one true “best” antioxidant (i.e., there is no master antioxidant ring to rule them all).


**Cheat code 5. How to measure lipid peroxidation in human samples**


Pioneering work on lipid peroxidation (see Figure 5), measured by expired pentane gas, is widely credited with founding the exercise redox biochemistry field [149]. Decades later how to measure lipid peroxidation, however, remains a vexing question. The vexing nature stems from the biochemical complex lipid peroxidation process generating manifold primary products, which are then metabolised to a myriad of secondary and tertiary products. While many context-specific factors influence what the “best” approach is, we suggest several approaches to measure lipid peroxidation in human samples (see Table 3).

**Figure 5 antioxidants-13-00877-f005:**
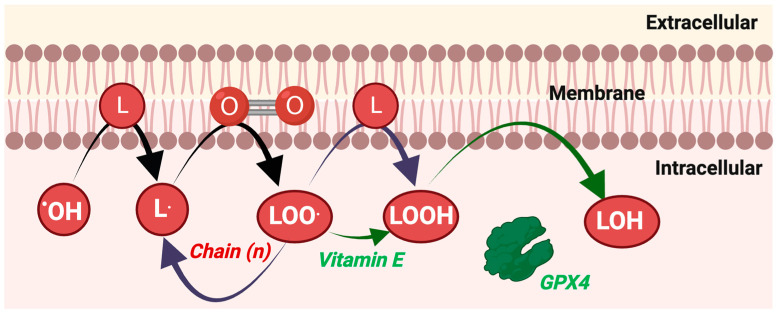
Start: Specific ROS, i.e., certain free radicals like hydroxyl radical or protonated superoxide [150], react with unsaturated lipids to generate a lipid radical (L·). Chain: L· rapidly reacts with oxygen to form peroxyl radicals (LOO^.^). LOO^.^ can react with another lipid to reform L· and produce LOOH. *This is an example of free radicals reacting fast together: oxygen and L*· *combine to form a non-radical (LOOH) and a free radical (L·)*. End: A chain-breaking antioxidant, like ubiquinol, vitamin E, or even certain persulphides [151], breaks the chain by intercepting LOO^.^. Enzymes metabolise the resultant LOOH species, most notably glutathione peroxidase 4 (GPX4) [152], usually converting them into LOH species.

**Table 3 antioxidants-13-00877-t003:** Methodological approaches for measuring lipid peroxidation in human samples.

*n*	Name	Description
**1**	**Lipidomics**	The discovery of ferroptosis—lipid peroxidation and iron-dependent cell death [153,154,155]—spurred interest in sophisticated technologies for measuring the myriads of oxidised lipid products at scale by using mass spectrometry (MS) [156]. One can even tell apart enzymatic from non-enzymatic F_2_-isoprostanes [157,158]. Commercial lipidomic analytical services are available.
**2**	**HPLC**	HPLC can measure lipid peroxidation products [31], such as F_2_-isoprostanes and MDA. In particular, the HPLC analysis of MDA is a valid and highly sensitive biomarker of exercise-induced lipid peroxidation [159].
**3**	**ELISA**	ELISA kits can measure some lipid peroxidation products [160], notably F_2_-isoprostanes. While they can be insufficiently specific in some cases [161], they have illuminated individual responses to exercise in humans [139,162,163,164].
**4**	**LOOH**	The ferrous oxidation−xylenol orange (FOX) assay developed by Wolff [165] can measure lipid hydroperoxides (LOOH). It can be combined with assays for lipid soluble antioxidants, such as vitamin E [166]. Specificity concerns can plague the FOX assay when certain types of highly oxidised lipids are analysed [167].
**5**	**Immunoblot**	Some lipid peroxidation products, such as 4-hydrononeneal (4-HNE), react with DNA and proteins [168]. Antibodies recognising 4-HNE-conjugated protein epitopes, formed secondary to Michael addition reactions, are available [169]. The global immunoblots can infer lipid peroxidation in human samples.
**6**	**Targeted approach**	In a variant of 5, a specific protein can be immunopurified for targeted analysis of protein-specific lipid peroxidation [170,171]. For example, by measuring a mitochondrial protein located near a membrane, this approach can assess organelle-specific lipid peroxidation.


**Cheat code 6. How to measure protein oxidation in human samples**


Numbers illustrate the formidable challenges of measuring protein oxidation per se, let alone in human samples. A typical cell contains about 50% (~10,000) of the proteins encoded by the human genome [172,173,174]. The susceptibility of virtually every proteogenic amino acid to oxidation [175] can generate many, over 10^10^, redox proteoforms [176]. A proteoform specifies different molecular forms of the protein inclusive of post-translational modifications [177,178]. Proteoform copy numbers can vary by ~10^6^. Such numbers make it exceedingly difficult to identify and quantify billions of possible redox proteoforms [179]. While many amino acids can be oxidised (cheat codes 8 and 9 consider cysteine), most studies measure modified forms of tyrosine—3-nitrotyrosine (3-NT)—and protein carbonylation, which affects multiple amino acids, such as lysine residues [59,180,181]. To measure protein oxidation in human samples, we suggest several approaches (see Table 4).

**Table 4 antioxidants-13-00877-t004:** Methodological approaches for measuring protein oxidation in human samples.

*n*	Name	Description
**1**	**Proteomics**	One can collaborate with specialist labs or access services to identify and quantify specific types of oxidised amino acid, such as carbonylated proteins (see Figure 6), on a proteome-wide scale by using bottom-up MS [182]. Sophisticated modification-specific workflows are available [183,184,185,186,187,188].
**2**	**ELISA**	Simple and user-friendly ELISA kits can quantify total protein carbonylation [189].
**3**	**Immunoblot**	Pan-proteome immunoblots with modification-specific antibodies or derivatisation reagents can be performed [190]. OxyBlot™ for protein carbonylation represents an enduringly popular approach [191].
**4**	**Fluorescence**	Derivatising protein carbonyl groups with fluorophores, such as rhodamine-B hydrazine, allows for quantifying their levels via SDS-PAGE [192,193], especially when protein content can be normalised with a spectrally distinct amine-reactive probe like AlexaFluor™647-*N*-hydroxysuccinimide (F-NHS). Novel N-terminal reactive reagents, such as 2-pyridinecarboxyaldehyde, may also be used [194,195].
**5**	**Targeted approach**	Specific proteins can be analysed by 1–4, such as MS for residue level analysis [196], when a protein is immunopurified. Targeted approaches can address specific questions [197,198], especially when the functional impact of the oxidation event is known (see Figure 7). For example, tyrosine 34 nitration impairs manganese SOD activity by electrically repelling superoxide [199]. Electrostatic repulsion helps explain why the rate of superoxide dismutation via O_2_·^−^ + O_2_·^−^ colliding to form hydrogen peroxide and oxygen is near zero [200]. Approach 4 combined with an ELISA assay may support protein-specific oxidative damage analysis in a microplate.

**Figure 6 antioxidants-13-00877-f006:**
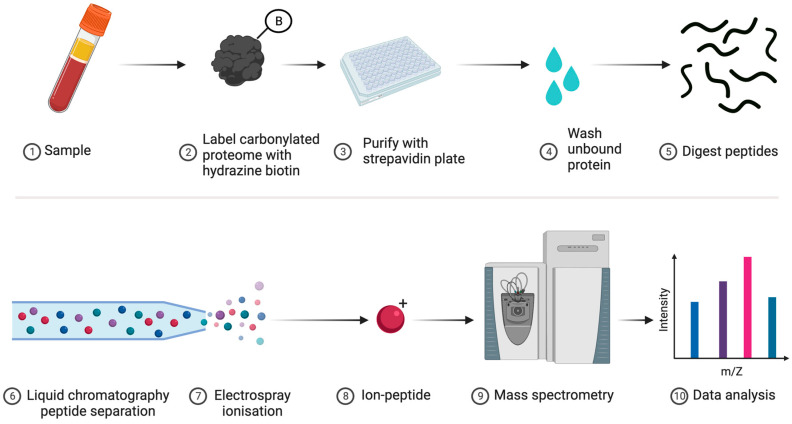
Proteomic-based analysis of carbonylated proteins. **1**. A biological sample is obtained. **2**. Carbonylated proteins are labelled with reducing agent cleavable hydrazine-conjugated biotin. **3**. The carbonylated proteome is separated from the rest of the sample via a streptavidin solid support, such as a 96-well microplate or superparamagnetic beads. **4**. Unbound protein is removed by extensive washing. **5**. After releasing the purified proteins by using a reducing agent, trypsin is used to digest the bound proteins into peptides. **6**. HPLC separates the peptides by their hydrophobicity before electrospray ionisation (**7**, **8**) and subsequent mass spectrometry analysis (**9**, **10**). In principle, this approach can determine the identity of the oxidised proteins and the amino acid residues that are oxidised.

**Figure 7 antioxidants-13-00877-f007:**
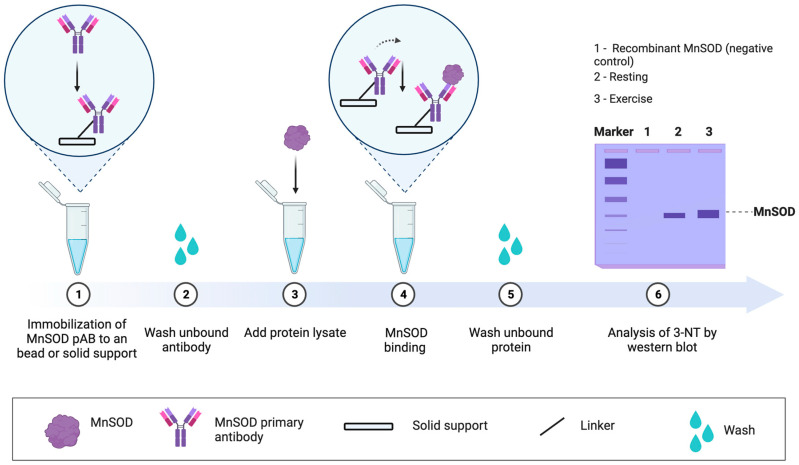
Analysis of MnSOD protein nitration using a targeted immunological approach. **1**. A MnSOD capture antibody is bound to protein A or G-coated superparamagnetic beads. **2**. The unbound antibody is removed by washing. **3**. A protein lysate containing MnSOD is added. **4**. MnSOD binds to the capture antibody. **5**. Unbound proteins are removed by extensive washing. **6**. MnSOD nitration is assessed by immunoblotting using a 3-NT antibody. Recombinant SOD can be loaded as a negative control. The basic approach described above can be adapted for other proteins or inverted. In the latter, a 3-NT capture antibody is used as bait to determine whether it binds MnSOD.


**Cheat code 7. How to measure DNA and RNA oxidation in human samples**


DNA lesions occur at a rate of 10,000 to 1,000,000 molecular lesions per cell per day and, if left unrepaired, can block DNA replication and transcription or lead to other serious genome aberrations [201]. Several excellent assays exist with the capability of capturing either DNA strand breaks (single or double stranded) or DNA oxidation (nucleoside base damage), and these techniques encompass molecular, fluorescence, chemiluminescence, analytical, and sequencing approaches (see Table 5).

Cellular RNA damage is far more abundant than DNA damage; however, only a few in vivo RNA oxidation indices have been reported. The primary by-product of RNA oxidation, 8-oxoGuo, is chemically similar to 8-oxodG, and this can cause interpretational issues if the correct analysis tool and sample type are insufficiently considered [215]. The principles for analysis of oxidised RNA are based on:**Assay techniques:** The validated analytical tools relating to DNA oxidation are applicable to RNA oxidation, as the latter is quantified by using ELISA, PCR-based technology, and chromatograph procedures, such as HPLC with electrochemical potential detection [216] and liquid chromatography/mass spectrometry or gas chromatography/mass spectrometry [217,218]. Note that most ELISA kits cannot discriminate between RNA and DNA oxidation products.**Sample type**: RNA oxidation can be quantified in urine, blood, and/or tissue (cells). The detection of 8-oxoGuo urinary excretion is possible but must be corrected for urine dilution (via urine volume, creatinine, or density). Blood plasma is an acceptable material to measure RNA oxidation with, although the data should be carefully interpreted with appropriate physiological modelling [217,219]. Tissue quantification has the advantage of tissue-specific interpretation, unlike plasma and urine collection.


**Cheat code 8. How to measure redox regulation in human samples**


While the redox regulation (see Box 1) literature dates back to at least 1966 [220], with reviews on the subject published in the 1980s [221], the turn of the millennium [222], marked a seismic shift in how the field interpreted exercise-induced oxidative stress [223]. Building on early work [224,225,226], landmark studies demonstrated that nutritional antioxidants impaired beneficial molecular adaptations to exercise in humans [227,228,229]. To measure redox regulation in humans [230,231,232], we suggest multiple approaches (see Table 6).

Box 1A brief overview of the redox regulation concept.**Redox regulation** defines the process whereby ROS-sensitive protein- and cysteine-residue-specific redox changes regulate protein function [30,64,65]. A paradigmatic example originating in the 1990s concerns the tyrosine phosphatase PTP1B [233]. The active site cysteine, Cys215, of PTP1B acts as a nucleophile—a chemical species that forms bonds by donating an electron pair—to dephosphorylate protein tyrosine. To act as a nucleophile, the sulphur atom in Cys215 must be in a reduced, deprotonated state (i.e., RS^−^). ROS can inactivate PTP1B by disabling nucleophilic catalysis [234]. For example, hydrogen peroxide can react with Cys215 to form a sulphenic acid. As an electrophile—a biochemical species, such as a Lewis acid, that seeks out electron pairs for bonding—the sulphenic acid prevents the cysteine from launching a nucleophilic attack on a phosphate bond [66,235]. Reviews on the mechanisms, biological function, and open questions in the redox regulation field are available elsewhere [30,64,65,236,237,238,239,240,241].

**Table 6 antioxidants-13-00877-t006:** Methodological approaches for measuring redox regulation in humans.

*n*	Name	Description
**1**	**Proteomics**	Sophisticated MS approaches, such as cysteine-reactive phosphate tag technology [242], can identify and quantify cysteine oxidation with residue resolution on a proteome-wide scale [243,244,245,246,247]. Modern methods allow for deep coverage of the cysteine proteome, such as ~34,000 residues across ~9000 proteins [242]. For reference, the full human proteome contains over 200,000 cysteine residues distributed across over 18,000 proteins [248]. New advances provide deep coverage and better quantification [249,250]. Protein-targeted methods are also possible [251,252]. Proteomic services, with some including data analytical packages, are available.
**2**	**Immunological approach**	Approaches (see Figure 8) include immunocapture before the streptavidin immunoblotting of biotin-conjugated oxidised cysteines for target-specific cysteine oxidation [253]. Non-reducing immunoblotting quantifies the oxidation of some proteins, such as protein kinase G [254]. For the many proteins that fail to exhibit endogenous oxidation-induced mobility shifts, cysteines can be derivatised with mobility-shifting polyethylene glycol (PEG) payloads [255,256,257,258,259,260]. These assays quantify cysteine redox proteoforms [261].
**3**	**Outcomes**	Some assays can infer the outcomes of redox regulation without measuring cysteine oxidation [262,263,264]. Transcriptional approaches, such as qPCR analysis, can infer the activation of redox-sensitive gene expression programmes, notably Nrf2 * activity. Immunoblot approaches include (1) degradation of KEAP1 * to infer Nrf2 activity; (2) monitoring a reporter, such as the phosphorylation of a signalling protein; and (3) quantifying protein content, such as heat shock proteins [265]. Changes in antioxidant enzyme activity, glutathione, or oxidative damage may also be instructive.

* Discovered in 1994 [266], Nrf2 is a transcription factor responsible for inducing the expression of over 300 genes, with many of them being linked to antioxidant defence, such as the gene encoding the rate-limiting step in glutathione biosynthesis—gamma glutamate cysteine ligase subunits [267]. Although the mechanisms are complex, Nrf2 is inhibited in the cytosol by KEAP1. The cysteine oxidation of KEAP1 releases Nrf2 to translocate to the nucleus [268].

**Figure 8 antioxidants-13-00877-f008:**
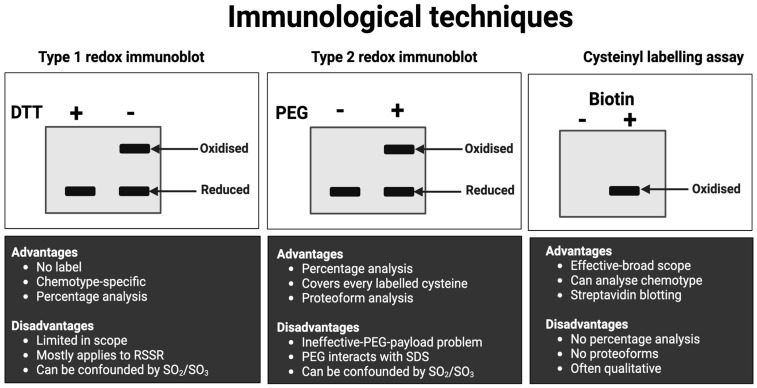
A visual overview of the main macroscale (slab-gel formatted) immunological techniques for measuring protein-specific cysteine redox state in human samples. See Table 6 for specific details. Their main advantages and disadvantages are listed for reference. Abbreviations: DTT = 1,4-dithiothreitol, PEG = polyethylene glycol, RSSR = disulphide bond; SO_2_ = cysteine sulphinic acid; SO_3_ = cystine sulphonic acid, and SDS = sodium dodecyl sulphate.

Selected “do” and “don’t” guidelines for measuring redox regulation in human samples include:Do minimise artificial cysteine oxidation by alkylating reduced cysteines with suitable reagents, such as *N*-ethylmaleimide (NEM). Note that these reagents can label other amino acids and react with sulphenic acids and persulphides [144,269,270].Do bear in mind that not every protein and every cysteine are yet measurable in one run by using MS technology [271].Do consider that different workflows measure different forms of cysteine oxidation, the so-called chemotypes. For example, a chemotype-specific proteomic approach demonstrated that fatiguing exercise increased S-glutathionylation, i.e., cysteine covalently attached to glutathione via a disulphide bond, in mice [272].Do consider supplementing the analysis with global readouts of cysteine oxidation, such as Ellman’s test [273], or chemotype-specific pan-proteome immunoblots [274].Do consider that many techniques do not measure “over-oxidised” chemotypes, such as sulphinic acids.Do not assume that a technique will necessarily work! Mobility-shift immunoblots usually fail to detect the protein because the bulky PEG payloads sterically block primary antibody binding.Do not assume cysteine oxidation is functional without evidence.Do not assume the cysteine oxidation is necessarily oxidative eustress without evidence.Do not assume an outcome assay result is caused by cysteine oxidation without evidence.


**Cheat code 9. How to use redox ELISA technology to measure protein cysteine oxidation in humans**


Simple and effective ELISA-based fluorescent immunoassays can measure target-specific cysteine oxidation, such as the antibody-linked oxi-state assay (ALISA) and RedoxiFluor [275,276]. They use redox state encoded fluorophores, such as AlexaFluor™647-C_2_-maleimide (F-MAL), to quantify the cysteine oxidation of an immunopurified protein in a microplate (see Figure 9). ALISA/RedoxiFluor bring the benefits of the microplate format, such as high-throughput analysis, to the redox regulation field (see Table 7). They open up new and exciting “off-the-shelf” opportunities for measuring redox regulation in human samples.

ALISA and RedoxiFluor can shed light on exercise- and nutrient-sensitive cysteine oxidation [277]. For instance, ALISA revealed how acute maximal exercise decreased cysteine oxidation of the catalytic core subunit of the serine/threonine PP2A protein phosphatase in human erythrocytes [278]. A follow-up benchmarking study demonstrated excellent assay performance across multiple analytical metrics, from accuracy to reliability [279]. Further, maximal exercise increased the cysteine oxidation of the redox-regulated GAPDH protein [280]. Visually displayed orthogonal assays, such as gel-based ALISA [281], confirmed the validity and specificity of the “unseen” microplate data. Selected points relevant to applying redox ELISA technologies in human samples include:Standard operating procedures are available [276]. The cysteine-labelling procedures can be adapted to suit specific experimental needs. For example, to omit some costly preparatory steps, reduced cysteines can be labelled with an F-MAL in ALISA. In this case, increased cysteine oxidation would decrease the observed F-MAL signal.The assays can operate in different modes, from global (i.e., all proteins/no antibodies) to multiparametric array mode, in microscale and macroscale (e.g., slab-gel format).In some cases, the assays provide information on protein function, such as the difference in transcription factor cysteine redox states in the cytosol vs. the nucleus.Interpretationally, a change reflects a difference in the rate of ROS-sensitive cysteine oxidation and antioxidant-sensitive reduction across the entire protein. The summed weighted mean of all individual residues responding to both ROS and AOX inputs is useful.The assays are compatible with chemotype-specific* cysteine labelling [282] and direct-reactivity approaches [283]. For example, the methods are compatible with reaction-based sulphenic acid fluorophores [284].Unless the target has one cysteine like ND3-Cys39 in complex I [285,286], the assays cannot disclose the identity of the oxidised cysteine residues.For some exercise-sensitive proteins, such as PGC-1α [287,288,289], the lack of an ELISA kit may make it impossible to run the assay.

*Relevant to cheat codes 8–9, the specificity of some chemotype approaches is actively debated [290,291,292]. For example, vitamin C can reduce S-nitrosated cysteine residues, but it also reacts with other chemotypes, such as sulphenic acids [293].


**Cheat code 10: How to exploit mathematical modelling and computational analyses in redox biology**


So far, we have presented cheat codes for measuring oxidative stress by using experimental methods in humans. However, some questions in biology cannot be tackled by using experimental methods alone and may require the aid of theoretical approaches. Systems biology investigates the dynamic properties and interactions within a biological object, at cellular and/or organismal level, in a qualitative and quantitative manner by combining experimental and mathematical approaches [294,295]. The iterative cycle of data-driven modelling and model-driven experimentation, in which hypotheses are formulated and refined until they are validated, can elucidate the emergent properties and mechanisms governing cellular processes and higher-level phenomena [296,297]. Such analyses are used in redox biology to quantitatively characterise redox processes and link them with physiological outputs [298]. For a comprehensive review, the reader is referred elsewhere [240,299,300,301,302,303]

Here, we highlight quantitative studies that have interesting implications for the production, metabolism, and signalling properties of specific ROS [49,304,305,306,307,308,309,310,311,312]. Antunes and Brito formulated a minimal model of hydrogen peroxide [313] and provided simple equations that can be used in combination with experimental measurements to estimate the response time and oxidation profile of cysteine-based redox switches, providing insights into the mechanisms of redox signalling. Pillay and colleagues fitted experimental data in computational models and performed a supply–demand analysis for hydrogen peroxide, that is, its production and consumption [314]. Their analysis showed that the activities of enzymes responsible for hydrogen peroxide consumption can act synergistically in the face of increasing hydrogen peroxide supply, to limit its steady-state concentration. A recent computational study investigated how far superoxide and hydrogen peroxide can travel through capillaries, arterioles, and arteries; what concentrations can be attained under different conditions; and the main determinants of these distances and concentrations [315]. A finding relevant to measuring oxidative stress in blood is that 36% and 82% of the plasma hydrogen peroxide is absorbed by the erythrocytes in the capillaries and arterioles, respectively. In another compelling example, it is commonly assumed that neutrophils can directly oxidise bacteria and other cell structures via a short burst of superoxide and hydrogen peroxide production. However, Winterbourn et al., by modelling the reactions of superoxide and myeloperoxidase, investigated the fate of superoxide and hydrogen peroxide in the neutrophil phagosome and provided an alternative explanation where superoxide acts as a (i) cofactor for hypochlorous acid production by myeloperoxidase and/or as a (ii) substrate for the production of other ROS [44,316].

Simple back-of-the-envelope calculations can yield important quantitative insights into biological processes/mechanisms and provide experimentally testable predictions [317,318]. Such calculations may often include simple order-of-magnitude estimates and algebraic equations. A theoretical analysis using only kinetic rates revealed that vitamin C and vitamin E are unlikely to have a major impact on exercise-induced redox signalling by scavenging hydrogen peroxide [3]. By using the kinetics and concentrations of hydrogen peroxide, it has also been shown that the frequently reported increases in hydrogen peroxide after exercise are not sufficient to induce redox signalling, except for the case of micro-domains, such as subcellular compartments or organelles [319]. Finally, we used simple stoichiometric calculations to show that oxidative stress leading to metabolic reprogramming within erythrocytes controls the concentrations of key molecules, such as ATP, NADPH, and 2,3-bisphophoglycerate [320]. In sum, simple numerical calculations can help characterise biological phenomena quantitatively, providing a deeper understanding of a given redox system. Overall, the ten elaborate cheat codes, as summarised in Table 8, should provide an invaluable resource for measuring oxidative stress in humans.


**Part 3: Perspectives**



**Using cheat codes to measure oxidative stress in humans**


Unlike in mathematics, there is no universal ground-truth answer for selecting the most appropriate cheat code-stratified oxidative stress assay(s) to implement in human studies [4,321]. Among the many factors that may be accounted for (e.g., sampling time points [322]), the following are influential:
Whether the redox approach is general or targeted, where the general “catch-all” approach analyses focus on as many distinct oxidative stress processes as possible and the targeted ones focus on a specific process, such as lipid peroxidation. In both cases, multiple process-specific analytical indices are preferred. Still, the depth of the analyses depends on whether oxidative stress is a primary, secondary, or tertiary biochemical outcome variable.The type of biological material acquired, usually blood and/or tissue samples, and the number of samples dictate what can be measured and how. For example, performing MS-based proteomic analysis on 100 samples is unlikely to be financially viable. Relatedly, the relevant equipment and expertise to undertake the analysis must be available.Whether (1) a redox-active molecule (such as antioxidant or pro-oxidant [323]) is being studied, (2) oxidative eustress or distress (e.g., harmful age-related DNA damage [324]) is being studied, and (3) oxidative stress is linked to a given outcome variable, such as exercise adaptations.

We have devised a research question-grounded decision tree for selecting the most appropriate cheat code(s) to implement (see Figure 10). For example, answering the question “*are you interested in oxidative damage or redox regulation?*” with “*oxidative damage*” simplifies the decision-making process by omitting cheat codes 8 and 9 at the outset. From this starting point, answering other questions should prove instructive for selecting suitable cheat code-stratified assays, for example, selecting indices of mitochondrial lipid peroxidation, such as organelle-specific 4-HNE immunoblotting as per cheat code 5, when using a lipophilic mitochondria-targeted antioxidant to modulate exercise-induced oxidative stress [325,326,327].


**Getting the NAC of it: A worked example**


To demonstrate how to use the cheat codes, we define a hypothetical prospective example concerning whether NAC acts as an antioxidant to improve exercise performance [328] by reversing ROS-induced fatigue [329]. In this example, an ROS input (the exercise-induced redox signature) produces a fatigue output via a process (redox mechanism) involving oxidative damage to contractile proteins, such as myosin chain isoforms. Based on our answers to the questions in the decision tree (see Table 9), we might use the following cheat codes:Cheat code 4 to verify increased NAC loading via HPLC-based analysis of plasma NAC and assay GSH and GPX activity in erythrocytes or tissue lysates to infer whether NAC supports the glutathione redox system [330]. This might be expected to alter peroxide metabolism and hence oxidative damage to proteins via lipid peroxidation products, such as 4-HNE [168].Cheat code 5 to measure lipid peroxidation. In plasma, one might measure LOOH and 4-HNE via the FOX assay and immunoblotting, respectively. In tissue samples, one might measure 4-HNE via immunoblotting. Equally, one might implement a F_2_-isoprostanes ELISA in plasma or tissue [331].Cheat code 6 to measure oxidative damage to contractile proteins by using targeted analysis and the immunocapture of a specific protein followed by immunoblot analysis for 4-HNE, 3-NT, or protein carbonyls [197]. Like how cutting fingernails yields thiyl radicals in alpha keratin [332], mechanical stress produces protein-based free radicals [333]. Hence, one could add a spin trap to “clamp” protein radicals for targeted immunoblot analysis with an anti-trap reagent [334]. If only circulating samples were available, the same approaches could be applied in these compartments to test the plausibility of NAC minimising oxidative damage to proteins (albeit non-contractile ones).Cheat codes 8–9 with chemotype analysis to determine whether NAC, by supporting hydrogen sulphide production, elicits beneficial effects by inducing contractile protein-specific persulphidation [335,336,337]. If fluorescent labels are used, then cheat codes 6, 8, and 9 could be implemented simultaneously [338], for example, gel-based detection of persulphidation before 4-HNE immunoblotting.

So far, we have elaborated our decisions based on the financial means to investigate two distinct modes of action in a biochemistry-guided catch-all approach. If financial or related concerns like time were limiting, the essential codes, corresponding to a minimal yet mechanistically cohesive approach, would be:Cheat code 4: GSH levels (systemic or tissue). Or cheat code 10 (see below).Cheat code 5: 4-HNE blot (systemic or tissue).Cheat code 6: myosin-specific 4-HNE levels (tissue).

To appreciate the benefits of the cheat code guide, it is worth contrasting even the minimal analysis to what might otherwise be performed. Without the guide, one might easily elect to measure glutathione concentrations in plasma, lipid peroxidation via TBARS, or 4-HNE levels globally as the incorrect conceptual/methodological analogues to cheat codes 4–6. It might also be speciously contended that NAC scavenges ROS. The kinetic plausibility of this hypothesis for key ROS like superoxide and hydrogen peroxide is questionable [336]. Even if it were kinetically plausible, the products must be considered. For example, the NAC thiyl radical can generate superoxide! Hence, an alternative analytical selection would confound the mechanistic interpretation of the study, constraining the ability of the researchers to investigate their hypothesis.

Interpretationally, the example investigated the potential existence of a causal relationship between an exercise-induced redox input, affected by specific ROS and AOX, and a whole-body output—exercise performance—via a skeletal muscle fatigue process mediated, in part, by the oxidation of contractile proteins. If we assume that the fatigue process mechanism is correct, then for NAC to exert a beneficial effect, in this way, three cheat code checkable points hold:NAC enters the circulation before it or a metabolite thereof accumulates in skeletal muscle (checked via cheat code 4: HPLC analysis of NAC).NAC indirectly acts as an antioxidant by impacting a process that influences the oxidation of contractile proteins. The former can be checked via GSH-related lipid peroxidation analysis (cheat codes 4–5) and the latter by targeted oxidative damage analysis pursuant to cheat code 6 or hydrogen sulphide donor effect as per cheat code 8 or 9.By so doing, NAC impacts a whole-body marker of fatigue such as exercise performance.

How one interprets oxidative stress depends on the selected outcome variable. In this case, interpreting oxidative stress analyses is conditional on a non-redox outcome: exercise performance. The same redox evidence could be interpreted differently [77] as delaying fatigue to perform more work delivers an immediate performance benefit but may sacrifice exercise adaptations in the long run.


**Getting a NAC for the numbers**


After focusing on NAC and glutathione, we provide a simple quantitative example on glutathione synthesis to showcase how to implement cheat code 10. We used an equation estimating glutathione synthesis: Glutathione synthesis = k_synthesis_ × [L-cysteine] × [L-glutamate]. It models the process as a second-order reaction and uses the intracellular concentrations of L-cysteine and L-glutamate, as well as the rate constant of de novo synthesis [339]. The rate constant and concentrations data were extracted from the literature [339,340]. We formed the model and performed all calculations by using the R Statistical Software and RStudio. By using this model, we calculated glutathione synthesis under two different nutritional scenarios: (i) NAC supplementation and (ii) dietary L-cysteine deficiency. Under physiological conditions, erythrocyte glutathione synthesis was estimated at ≈700 μM per day, corroborating previous experimental findings [341]. Supplementation with NAC has been shown to lead to a 2-fold increase on average in L-cysteine concentration for about 4 h [342]. Based on this, NAC supplementation was estimated to increase glutathione synthesis by ≈16%, leading to ≈816 μM per day. Dietary L-cysteine deficiency can decrease intracellular cysteine concentration by ≈55%. Lower L-cysteine availability was estimated to decrease glutathione synthesis by ≈40%, leading to ≈420 μM per day. The lower glutathione synthesis due to L-cysteine deficiency is supported by experimental data. This simple and quick quantitative example demonstrates how simple calculations can provide biologically meaningful and experimentally falsifiable/testable estimations. Our example is available on GitHub (https://github.com/PanosChatzi/Quantitative-redox-biology-calculating-glutathione-synthesis).


**On the future of translational human redox research**


The failure to measure oxidative stress obscured our understanding of whether nutritional antioxidants alter the onset of disease in humans [32], which is now admittedly a fanciful concept that can be likened to appealing to magic [343]. This pressing problem leached into most fields, especially exercise redox biochemistry. Many studies investigating whether nutritional antioxidants blunt exercise adaptations failed to measure oxidative stress [3]. Our cheat codes and decision selection tool can prevent these problems from reoccurring by enabling researchers to measure oxidative stress in humans. They bring a range of techniques to bear on the elucidation of the roles that ROS play in humans, notably in the progression of diseases and their potential amelioration through diet and exercise. Focusing on diet and exercise themselves, instead of prophylactic antioxidants that often lack the biochemical capacity to modulate the relevant redox reactions, defines a promising direction for advancing current knowledge. Halliwell counted diet and exercise amongst the best strategies to manipulate redox biology, such as ROS levels, in humans [36].

The future of oxidative stress research is bright. The cheat codes can shed light on fundamental redox phenomena. Recent data link cardon dioxide/bicarbonate-derived species to direct protein cysteine oxidation [344,345,346]. In effect, cardon dioxide/bicarbonate accelerate the process. The practice of using bicarbonate to modulate performance may illuminate cysteine oxidation mechanisms in humans. Skeletal muscle studies are ideally placed to pioneer single cell oxidative stress research, especially when clear fibre-type differences [347,348] make such analysis phenotypically meaningful [349]. They can also address lingering questions in redox signalling [350,351,352], particularly around the spatial regulation of the process with the potential for phase transition effects [353,354]. Finally, the field is set to play a leading role in unravelling the addressing individuality at the dawn of the personalised redox biology era [355]. For example, potential differences in how exercise-induced oxidative stress manifests in males compared with females remain largely unexplored.

Returning to the “at least for now” clause in cheat code 1, sophisticated tools, such as MitoNeoD for measuring mitochondrial superoxide levels [96], offer possibilities for eventually measuring ROS levels in vivo by using non-invasive technologies, such as positron emission tomography (PET) [356]. Notable developments relate to the PET tracing of established ROS-reporters like ethidium in animal models [357]. While there are many challenges to overcome and, even then, limitations will still apply (e.g., selectivity to one specific ROS [89,358]), future innovations may enable the non-invasive measurement of ROS levels in humans. More broadly, we expect the cheat codes to evolve in line with technical advances. For example, and with reference to cheat code 5, a promising recent approach leveraged an enzymatic conjugation strategy to identify and quantify novel F_2_-isoprostanes in human urine [359]. Novel analytical approaches, such as the drop blot for single-cell Western blot analysis [360,361], and artificial intelligence breakthroughs allowing for the production of designer proteins, notably antibodies [362,363], provide fertile ground for further advances, for example, a designer enzyme to selectively reduce sulphenic acids over other chemotypes.

The sheer biochemical complexity of oxidative stress presents multifaceted analytical challenges [72,364]. Like a redox Laplace demon, there are no means to measure all of the relevant biochemistry in humans—to view every pixel of the picture in every dimension from ROS to oxidative damage. For example, even in comparatively simple cellular systems, over 2000 human proteins have yet to be measured at the peptide level, let alone the proteoform level [365,366,367]. The addressable yet still formidable challenge concerns capturing as much of the oxidative stress picture as possible by using “omic” approaches, from state-of-the-art MS approaches to novel technologies, such as nanopore-based protein sequencing [368,369,370]. We call for a concerted community-wide effort to capture the multi-dimensional oxidative stress space in human samples to elucidate the biochemical nature of the phenomenon. We envisage the resources rationally guiding the validation of a large inventory of single or multiple biomarker panels comprising endogenous process-specific redox reporters, such as a biomarker of mitochondrial complex III-specific superoxide production [86,371]. They would enable one to rationally measure representative pixels as “snapshot” biomarkers of process-specific oxidative stress biochemistry.

## 2. Conclusions

We tackled the long-standing problem of how to navigate the appreciable complexities of measuring oxidative stress in humans. Our solution defined a cheat code-stratified suite of valid analytical approaches for measuring process-specific, such as lipid peroxidation, aspects of oxidative stress in humans. The cheat codes are poised to advance translational redox research by supporting the measurement of oxidative stress in humans. Much like how they demystify and simplify seemingly impossible video gaming challenges, a set of redox “cheat codes” can hopefully demystify the intricacies of oxidative stress research, breaking down complex concepts into manageable steps for starting and guiding the voyage of discovery that is measuring oxidative stress in humans.

## Figures and Tables

**Figure 1 antioxidants-13-00877-f001:**
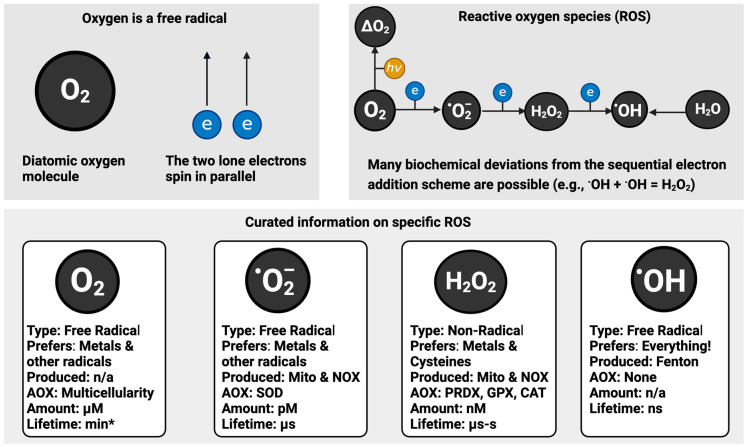
Oxygen (O_2_) and reactive oxygen species (ROS). The upper left panel depicts diatomic ground-state molecular dioxygen as a free radical with the two lone electrons (e) spinning in parallel according to the laws of quantum mechanics, such as the Pauli exclusion principle. The upper right panel displays the electronic relationship between different ROS. From O_2_ as a starting point, an input of energy can flip the spin whereby the lone electrons spin in parallel in singlet oxygen (^1^ΔgO_2_). Alternatively, the univalent reduction of O_2_ produces the superoxide O_2_·^−^. The proton-coupled univalent reduction of O_2_·^−^ produces hydrogen peroxide (H_2_O_2_). Without the coupling of protons, the extremely unstable peroxide dianion, a non-radical, would be produced. The univalent reduction of H_2_O_2_ yields a hydroxide anion (omitted for clarity) and the hydroxyl radical (·OH). In some cases, ·OH can be formed from water (H_2_O). For example, an input of energy sufficient to split the water in a homolytic fission reaction produces a hydrated electron, a proton, and ·OH. The lower panel visually displays curated information on specific ROS. Stating a concentration value for ·OH is difficult: its diffusion-controlled reactivity effectively prevents it from accumulating to an appreciable level. Abbreviations: NOX = NADPH oxidase, Mito = mitochondria, PRDX = peroxiredoxin, GPX = glutathione peroxide, and CAT = catalase.

**Figure 2 antioxidants-13-00877-f002:**
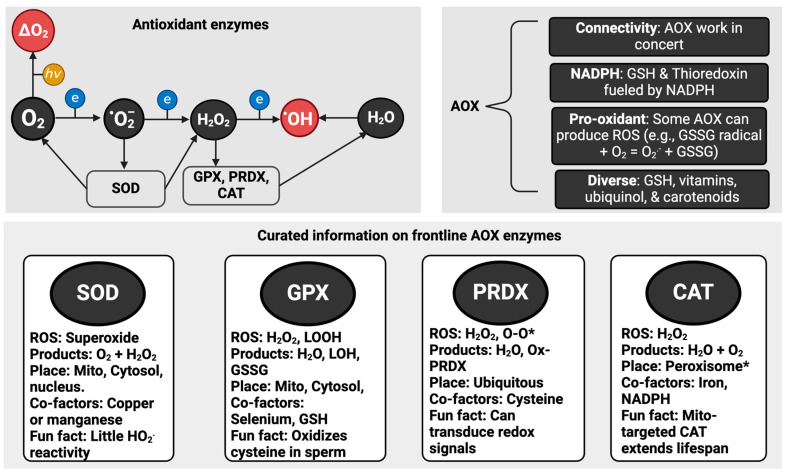
Antioxidants (AOX). The upper left panel visually displays the link between key reactive oxygen species (ROS) and enzymatic antioxidants. Superoxide dismutase (SOD) isoforms metabolise superoxide (O_2_·^−^) to oxygen (O_2_) and hydrogen peroxide (H_2_O_2_). Peroxiredoxin (PRDX) and glutathione peroxidase (GPX) isoforms can metabolise H_2_O_2_ to water (H_2_O). Catalase (CAT) effectively dismutates H_2_O_2_ to O_2_ and H_2_O. There are no known AOX enzymes in humans designed to specifically react with singlet oxygen (^1^ΔgO_2_) and the hydroxyl radical (·OH). The upper right panel visually displays interesting features of antioxidants, from the connectivity principle to their biochemical diversity (see the main text for details). The lower panel visually displays curated information for the key AOX enzymes. The PRDX isoforms can react with a range of species with an O-O bond, such as protein and lipid hydroperoxides. Abbreviations: GSH = reduced glutathione, GSSG = oxidised glutathione, GSSG radical = oxidised glutathione free radical, and Mito = Mitochondria.

**Figure 3 antioxidants-13-00877-f003:**
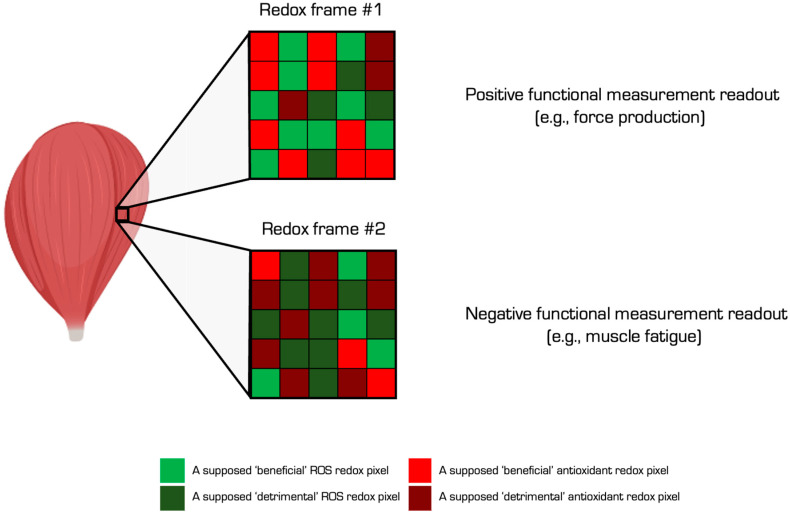
The oxidative stress biochemical pixel concept visualised. In this purely illustrative example, the two redox frames contain different redox pixels corresponding to specific ROS (green) and antioxidant (red) biochemical reactions that we have labelled, for the purposes of the figure, “beneficial” or “detrimental”. How we interpret the pixels depends on the measurement outcome. In this example, the biochemistry displayed in frame 1 corresponds to a positive functional readout: muscle force production. Conversely, the biochemistry displayed in frame 2 corresponds to a negative readout: muscle fatigue. The example also illustrates the dynamic nature of the biochemistry, where frame 1 could give way to frame 2 over time, which illustrates how the outcome also depends on when we look.

**Figure 4 antioxidants-13-00877-f004:**
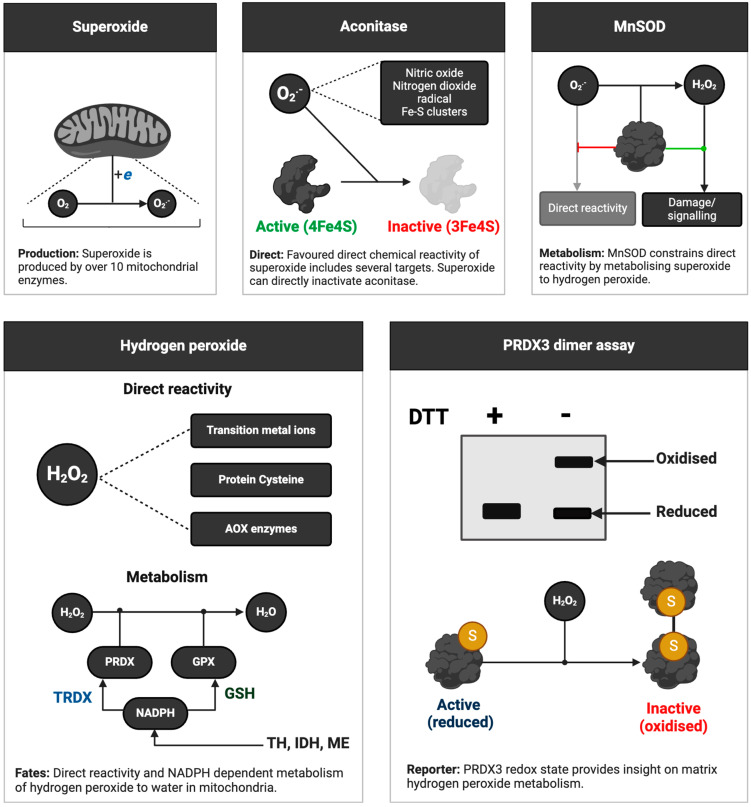
Inferring mitochondrial ROS levels in human samples. Several mitochondrial enzymes can catalyse the univalent reduction of oxygen to superoxide (O_2_·^−^). Superoxide levels in the mitochondrial matrix can be inferred by measuring the activity of the tricarboxylic acid cycle enzyme: aconitase. Superoxide directly and rapidly inactivates aconitase. Manganese superoxide dismutase (MnSOD) activity (see cheat code 4) dictates the concentration of matrix superoxide, playing an important role in controlling its direct reactions with other species, such as aconitase, and the production of hydrogen peroxide (H_2_O_2_). Hydrogen peroxide primarily reacts with antioxidant enzymes, such as PRDX, transition metal ions, and protein cysteine residues. The concentration of hydrogen peroxide in the mitochondrial matrix is mainly controlled by the activities of the NADPH-dependent glutathione (GSH) and thioredoxin (TRDX) redox systems, which are fuelled by the activity of transhydrogenase, isocitrate dehydrogenase, and the malic enzyme. To infer matrix hydrogen peroxide homeostasis, one can measure PRDX3-specific cysteine oxidation using the dimer assay, whereby the oxidised, disulphide-bonded homodimer migrates at a higher molecular weight than the reduced monomers in the absence of a reducing agent, such as 1-4-dithiothreitol (DTT).

**Figure 9 antioxidants-13-00877-f009:**
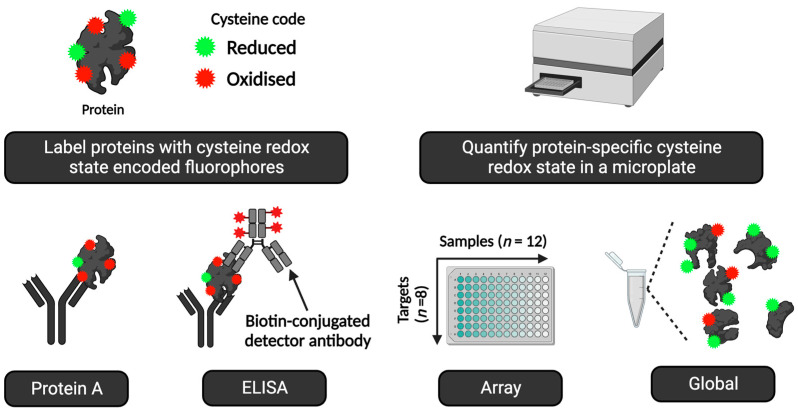
Visual overview of RedoxiFluor. In the upper left panel, reduced and oxidised cysteines in the sample are labelled with spectrally distinct fluorophores. The spectrally distinct fluorophores enable the cysteine redox state to be quantified in a number of ways in a microplate (upper right panel). As depicted in the bottom panel, four different assay variations can be used. From left to right: First, a protein A-coated microplate can be used to enrich a protein target of interest by using a suitable capture antibody. Second, a conventional ELISA comprising a capture and detector antibody pair can be used to quantify the relative target-specific cysteine redox state in percentage and moles. Third, either protein A or an ELISA mode can be deployed to measure the cysteine redox state of multiple targets in several samples in an array mode assay. Finally, the global cysteine redox state of the proteome can be determined in a microplate.

**Figure 10 antioxidants-13-00877-f010:**
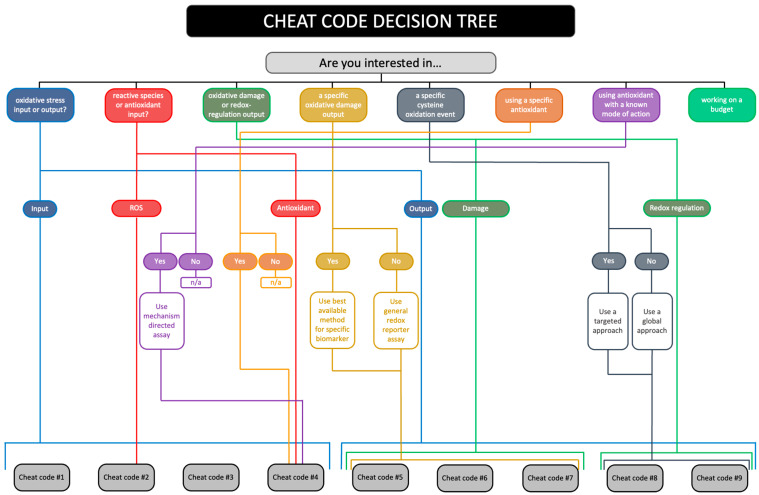
The research question-grounded decision tree for selecting the most appropriate cheat code(s) to implement for experimentally measuring oxidative stress in a prospective experimental human study. For example, if you are interested in a specific antioxidant, such as the mitochondria-targeted quinone (MitoQ), then you should consider implementing cheat code 4. By answering the other questions, one can obtain the most applicable cheat code(s). Note that cheat code 10 can be applied to all the answers and should be considered irrespective of the specificities of the study (i.e., it is widely applicable). For example, simple kinetic values for the reaction of MitoQ with different ROS would be instrumental for selecting appropriate mechanism-based assays. Please see the main text for a worked example.

**Table 1 antioxidants-13-00877-t001:** Suggested definitions of selected key terms.

Term	Definition	Example
** *Free radical* **	A molecule with one or more unpaired valence electrons that is capable of independent existence.	Hydroxyl radical
** *Non-radical* **	A spin-paired molecule devoid of unpaired electrons.	Hydrogen peroxide
** *ROS* **	Free radical and non-radical oxygen-derived reactive species.	Superoxide
** *Reactive species* **	A catch-all term for all free radical and non-radical reactive species that includes oxygen-, sulphur-, nitrogen-, and carbon-atom-centred species.	Peroxynitrite (sum of the protonated and deprotonated forms)
** *Reductant* **	A molecule that donates one or more electrons to a redox reaction partner.	NADPH donating a hydride to reduce oxidise glutathione reductase
** *Oxidant* **	A molecule that takes one or more electrons from a redox reaction partner.	Hydroxyl radical oxidising L-cysteine to a thiyl radical
** *Antioxidant* **	Any substance that delays, prevents, or removes oxidative damage to a target molecule. A substance that reacts with an oxidant to regulate its reactions with other targets, thus influencing redox-dependent biological signalling pathways and/or oxidative damage.	Vitamin E preventing lipid peroxidation by reducing an alkoxyl radical to a hydroperoxide CAT metabolising hydrogen peroxide to prevent reaction with other targets, such as a transition metal ion
** *Oxidative stress* **	An imbalance between ROS and antioxidants leading to disrupted redox regulation and/or oxidative damage.	See below
** *Oxidative eustress* **	A beneficial imbalance between ROS and antioxidants leading to disrupted redox regulation and/or oxidative damage.	Specific cysteine oxidation patterns and oxidative macromolecule damage in an acute-exercise context
** *Oxidative distress* **	A deleterious imbalance between ROS and antioxidants leading to disrupted redox regulation and/or oxidative damage.	Aberrant and potentially random patterns of cysteine oxidation and oxidative damage in a musculoskeletal ageing context

**Table 2 antioxidants-13-00877-t002:** Methodological approaches for measuring antioxidants in human samples.

*n*	Name	Description
**1**	**Enzyme activity**	The basis of modern SOD assays [127]—intercepting xanthine oxidase-induced superoxide before it reduces a reporter molecule like cytochrome C—was in place before SOD was discovered! [21]. Many valid microplate and gel-based SOD assays exist. They can be used to quantify SOD activity and, with appropriate controls, discern between isoforms [128,129,130]. Glutathione peroxidase (GPX) activity and CAT activity can be assessed by using established assays [31,131]. Meanwhile, PRDX and thioredoxin activities can be inferred via non-reducing immunoblotting. Assays to quantify thioredoxin reductase or glutathione reductase activity are available [132,133].
**2**	**Surrogate markers**	Antioxidant enzyme activity can be inferred by using surrogate markers [4]. For example, PRDX1 activity can reflect the phosphorylation state of the enzyme [134]. Other useful post-translational surrogates include manganese SOD acetylation state, disulphide bond formation in copper zinc SOD, and succinylation of GPX4 [135]. Measuring the concentration of a protein by immunoblotting or ELISA, in combination with other markers, can be used to infer a change in antioxidant activity [136].
**3**	**Glutathione**	High-performance liquid chromatography (HPLC)-based methods are available for measuring reduced (GSH), oxidised (GSSG), and total (GSH + GSSG) glutathione [37,137]. Although HPLC is preferred, plate- and kit-based assays are available. Controls (see cheat code 8) can prevent artificial cysteine oxidation [138].
**4**	**Low-molecular-weight molecules**	HPLC-based methods can measure the concentration of low-molecular-weight antioxidants like vitamin C [139]. EPR spectrometry can measure vitamin C and E radicals to provide information on their redox activity in human samples [140].

**Table 5 antioxidants-13-00877-t005:** Methodological approaches for measuring DNA damage in human samples.

*n*	Name	Description
**1**	**Fluorescence**	Single-cell gel electrophoresis (SCGE), or the comet assay, is a technically simple and sensitive method for quantifying single- and double-stranded DNA breaks in mononuclear cells. The SCGE assay embeds cells on agarose gel, and if supercoiling is relaxed due to a single or double break, the loop of the DNA is free to migrate during electrophoresis and thus form a comet tail [202]. The SCGE assay can resolve damage up to ~3 breaks per 109 Da [203]. Several variations in the SCGE assay exist. As oxidation of DNA can occur at a similar rate to strand breaks, base oxidation can be examined via a simple SCGE modification using lesion-specific enzymes (e.g., formamidopyrimidine DNA glycosylase) to accurately capture oxidised purines (guanine/adenine) and/or pyrimidines (thymine/cytosine). The assay may be combined with fluorescent in situ hybridisation (FISH) to detect whole-genome, telomeric and centrometric DNA, and gene region-specific DNA damage [202]. CometChip technology is now used to minimise sample-to-sample variation [204].
**2**	**ELISA**	ELISA application is a popular immunological method to measure DNA damage, mainly in the form of 8-oxodG. There are several commercially available kit-based options; however, caution is paramount regarding assay specificity and reliability.
**3**	**Molecular approach**	Polymerase chain reaction (PCR) or quantitative PCR (qPCR) can map nuclear and mitochondrial DNA damage at nucleotide resolution [205]. In PCR, DNA amplification is stalled at the damaged site via the blocking of the progression of Taq polymerase, which ultimately reduces the number of DNA templates that are devoid of any Taq-blocked lesions. qPCR quantifies DNA damage on both duplex strands. It is possible to quantitatively detect and analyse gene-specific DNA damage (and repair) by using qPCR and with only 1–2 ng of total genomic DNA. qPCR has caveats: it depends on high-molecular-weight DNA, well-defined qPCR conditions, and the intricate calculation of lesion frequencies [206,207]. Long-Amplicon Quantitative PCR (LA-qPCR) can also provide an overview of total genomic mitochondrial DNA damage [166]. Following double-stranded DNA damage, a repair response is usually initiated, where the subsequent phosphorylation of Serine-139 of histone H2AX ensues [208]. The γH2AX assay is relatively simple to execute and is based on immunofluorescence using a specific Serine-139-γH2AX antibody to show the location in the chromatin foci at the sites of DNA damage.
**4**	**Analytical approach**	HPLC coupled with tandem MS is reliable in detecting oxidative DNA damage (e.g., 8-hydroxy-2′-deoxyguanosine) with excellent separation of nucleosides. HPLC-MS yields robust information on the location of DNA damage, but high assay cost and required extensive experience can preclude assay use. Gas chromatography coupled with mass spectrometry (GC/MS) is highly sensitive to the detection of several forms of DNA damage, including those of the sugar moiety and four heterocyclic bases (e.g., 8-oxodG, 5-HMUra, 8-oxoAde, 5-OHUra, and 8-oxoGua) [209]. Although the technique provides impressive structural data in complex samples (such as the detection of a single DNA lesion in DNA with several lesions), the quantification of nucleoside forms of base damage is not as robust compared with liquid chromatography-based methods [210].
**5**	**Sequencing**	Innovative next-generation sequencing technology now exists, providing high-throughput and high accuracy DNA sequence data. RADAR-Seq [211], qDSB-Seq [212], and AP-Seq [213] are typical examples of sequencing-based technologies designed to quantify and map DNA damage on a genome-wide scale. This approach determines the precise gene locations of DNA damage. Interestingly, the quenching of fluorophores on account of the low redox potential of guanine had to be addressed before next-generation sequencing technologies could be developed [214].

**Table 7 antioxidants-13-00877-t007:** Type-stratified benefits of the ELISA formatted fluorescent immunoassays: ALISA and RedoxiFluor.

Type	Benefits	Description (Useful Properties as Applicable)
**ELISA**	Throughput Multiplexed Sensitive Rapid	High-sample n-plex analysis (adds statistical power) Parallel assessment of multiple 2–10 proteins (enables screening) Picomole sensitivity (supports human biomarker studies) Performed in 1 day with minimal hands-on time (benefits screens)
**Redox**	Cysteine holistic Percentages Moles Context Chemotype Process sensitive	Agnostic of any one cysteine residue (adds coverage of the entire molecule) Quantifies cysteine redox state in percentages (interpretationally useful) Quantifies cysteine redox state in moles (interpretationally useful) Provides cysteine proteome context (interpretationally useful) Supports chemotype-specific analysis (supports mechanistic studies) Results are sensitive to oxidative and antioxidative processes scaled across every cysteine residue on the target protein (interpretationally useful)
**Performance**	Valid Effective Accurate Reliable Reproducible Range	Draws on highly principled redox and immunological methods (robust) They work (e.g., compare to Click-PEG) (means to study the specific protein) Data correspond to ground-truth standards (adds percentage analysis) High consistency between samples (adds robustness) Delivers consistent results (adds robustness) Operates across a large dynamic range (useful for human applications)
**Microplate**	Simple Easy to perform Off the shelf Automated	Simple to understand, interpret, and operate (supports accessibility) Little technical skill required to deliver actionable results (accessibility) Compatible with commercial ELISA kits (accessibility) Delivers rapid and automated data within seconds (time efficient)

**Table 8 antioxidants-13-00877-t008:** Cheat code summary. The number and name of the codes are matched to analytical approaches were appropriate.

Code	Name	Analytical Approaches
1	Avoid the minefield of measuring ROS directly in humans (at least for now)	n/a
2	How to infer ROS production in human samples by using endogenous reporter molecules	Aconitase assay Peroxiredoxin dimer assay
3	How to hack “TAC” in human samples	TAC
4	How to measure antioxidants in human samples	Enzyme assays Surrogate markers Glutathione Low-molecular-weight compounds
5	How to measure lipid peroxidation in human samples	Lipidomics HPLC-MDA ELISA—F_2_-isoprostanes Fox assay—LOOH Immunoblot—4-HNE Targeted-protein-specific approach—4-HNE
6	How to measure protein oxidation in human samples	Proteomics Fluorescent-in-gel carbonylation assay ELISA—protein carbonylation Immunoblot—OxyBlot™ Targeted-protein-specific nitration
7	How to measure DNA and RNA oxidation in human samples	Fluorescence—comet assay Sequencing—RADAR-Seq Analytical approach—HPLC ELISA—8-oxo-G Molecular approach—QPCR
8	How to measure redox regulation in human samples	Redox proteomics Immunological assays, such as Click-PEG Outcomes, such as keap1 degradation
9	How to use redox ELISA technology to measure protein cysteine oxidation in humans	ALISA RedoxiFluor
10	How to exploit mathematical modelling and computational analyses in redox biology	Mathematical modelling and bioinformatics using appropriate software packages

**Table 9 antioxidants-13-00877-t009:** Worked example-specific answers to the research question-grounded cheat code(s) selection tool.

Are You Interested in…	Answer	Refinement	Selection Outcomes	Assay(s)
** *Oxidative stress input or output?* **	Output	n/a	Consider cheat codes 5–9 Disregard cheat codes 1 and 2	n/a
** *Reactive species or antioxidant input?* **	Yes	Interested in an antioxidant (yes)	Consider cheat code 4 Disregard cheat codes 1–3	n/a
** *Oxidative damage or redox regulation output?* **	Both	n/a	n/a	n/a
** *A specific oxidative damage output?* **	Yes	Yes	Consider cheat codes 5–6 Disregard cheat code 7 (based on mechanism)	Global 4-HNE immunoblot Contractile protein immunocapture for targeted 4-HNE analysis
** *A specific cysteine oxidation event?* **	Yes	Yes	Consider cheat codes 8 and 9	Gel-based analysis of persulphides using a fluorescent probe
** *Using a specific antioxidant?* **	Yes	n/a	Consider cheat code 4	n/a
** *Using an antioxidant with a known mode of action?* **	Yes	Use mechanism-directed assay	Consider cheat code 4	HPLC of [NAC]

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
