# Peer review of "Ten “Cheat Codes” for Measuring Oxidative Stress in Humans"

_antioxidants, 2024, doi:10.3390/antiox13070877_

Round 1

Reviewer 1 Report

The review article entitled "Ten “Cheat Codes” for Measuring Oxidative Stress in Humans" is well written and organized in a way that facilitates understanding of the topic. The focus of the review on the assessment of oxidative stress is extremely relevant and enriching to help outline the best experimental approach and select the most appropriate techniques with the most potential for the specific objective defined by the researcher. The document is excellent guidance for developing research in the area. The various topics are well explored and provide various advice and recommendations that are extremely important for the successful assessment of reactive oxygen species, antioxidants, biomarkers of oxidative damage and redox regulation.

Nothing to point.

Author Response

Dear Reviewer,

Thank you very much for your thoughtful and encouraging feedback on our review article entitled "Ten 'Cheat Codes' for Measuring Oxidative Stress in Humans." We are delighted to hear that you found the article well-written and organised in a way that enhances understanding of the topic.

We appreciate your acknowledgment of the relevance and enrichment our review brings to the assessment of oxidative stress. Our primary goal was to outline the best experimental approaches and to aid researchers in selecting the most suitable techniques for their specific objectives. It is gratifying to know that you consider our document excellent guidance for developing research in this area.

We are also pleased that you found the various topics well-explored and the advice and recommendations provided to be crucial for the successful assessment of reactive oxygen species, antioxidants, biomarkers of oxidative damage, and redox regulation.

Thank you once again for your positive review. Your feedback is invaluable and motivates us to continue our efforts in contributing to this important field of research.

Best regards,

James Cobley (on behalf of all the authors)

Reviewer 2 Report

No major comments

No detail comments

Author Response

Dear Reviewer,

Thank you for taking the time to review our manuscript entitled "Ten 'Cheat Codes' for Measuring Oxidative Stress in Humans." We appreciate your effort in evaluating our work.

Best regards

James Cobley (on behalf of all the other authors)

Reviewer 3 Report

Please include 10 codes at end of section 2.

Please summary all method methods match with the each code at end of section 2.

Please shorter text in the table, please put more columns in the table

please include references citation with numbers at end of column

Please change references to MDPI style

line 222

'3. Part 2. Ten “cheat codes” for measuring oxidative stress in humans'

There is part 2. It seems there is part 1

The table 8 is very good.

Can you please arrange other table similar like table 8 

The texts are very long in other tables.

Please shorter text in the table, please put more columns in the table

please include references citation with numbers at end of column

Please change references to MDPI style

 Figure 1,4,7

There are too many text in the frames so the text font is small. please reduce text in the figure.

Round 2

Reviewer 3 Report

The manuscript has been improved. The reference style and citations have been changed to MDPI style.

Some texts are very long in tables.

Figure legend style can be changed to large figure legend style (full wide of page)

 Text style

Please use Palatino Linotype through the manuscript also in citation and new table 8

Figure legend

Figure 4, 6, 7, 8 are large figure. please change to large figure legend style (full wide of page)

Table 1-7

The content of text are huge, please use large table (full wide of page)

Author Response

Thank you for reviewing our manuscript. In response:

  1. The text has been changed as requested.
  2. The legends have been changed as requested.
  3. The tables have been adjusted as requested. 

Round 3

Reviewer 3 Report

Author has improved the manuscript.

Please modify table 1 and 8.

changing Reference citation

Please include author contribution at end of manuscript. 

Table 1 and 8

Please span the wide of the column 3 and reduce  the wide of column 2.

Change table 8 font to Palatino Linotype

Reference citation should be Palatino Linotype. Please correct them through the text.

Author Contributions: For research articles with several authors, a short paragraph specifying their individual contributions must be provided. The following statements should be used “Conceptualization, X.X. and Y.Y.; methodology, X.X.; software, X.X.; validation, X.X., Y.Y. and Z.Z.; formal analysis, X.X.; investigation, X.X.; resources, X.X.; data curation, X.X.; writing—original draft preparation, X.X.; writing—review and editing, X.X.; visualization, X.X.; supervision, X.X.; project administration, X.X.; funding acquisition, Y.Y. All authors have read and agreed to the published version of the manuscript.”  Authorship must be limited to those who have contributed substantially to the work reported.

Author Response

Thank you. The changes have been made as instructed. 

All the best,

James